# The Effectiveness of a Poster Intervention on Hand Hygiene Practice and Compliance When Using Public Restrooms in a University Setting

**DOI:** 10.3390/ijerph16245036

**Published:** 2019-12-11

**Authors:** Aaron Lawson, Marie Vaganay-Miller

**Affiliations:** Belfast School of Architecture and the Built Environment, Ulster University, Newtownabbey BT37 0QB, UK; m.vaganaymiller@ulster.ac.uk

**Keywords:** Hand hygiene, behaviour, compliance, observation, intervention, poster, visual

## Abstract

Background: Most research on hand hygiene compliance in community settings indicates that compliance is poor. It is not conclusive as to whether poster interventions are effective at improving compliance. Methods: An independent, self-designed poster intervention was installed in one set of male and female public restrooms in a university campus in the UK. The hand hygiene practice and compliance of the university population was measured via indirect observation over a 60 day period. Results: During the pre-intervention observation period, 51.09% of the university population practiced basic hand hygiene compliance (washed hands with water, soap and dried afterwards), and 7.88% practiced adequate hand hygiene compliance (washed hands with water and soap for 20 s or more and dried afterwards for 20 s or more). During the post-intervention observation period, 55.39% of the university population were observed practicing basic hand hygiene compliance, and 7.97% practicing adequate hand hygiene compliance. Gender differences revealed that more females practiced basic hand hygiene in the post-intervention observation period (62.81%) than during the pre-intervention period (49.23%) and this was statistically significant (χ^2^ = 13.49, *p* = < 0.01). Discussion: The poster intervention had a limited effect on improving the basic and adequate hand hygiene compliance of the general population when using public restrooms. The use of independent, self-designed posters to improve hand hygiene practice and compliance is largely ineffective in the short term and should be used with caution in future intervention strategies.

## 1. Introduction

Most communicable diseases are transmitted via hands when they are not washed and dried properly or done so poorly [1]. It is estimated that the average person uses the toilet between five and eight times a day, and for people with chronic health conditions and disabilities, this may be more often [2]. Therefore, adequate hand hygiene practice and compliance at key times, such as after using the toilet, is critical in preventing the spread of communicable diseases in community settings [3]. 

Public restrooms which include any room or building with toilet facilities outside of the home environment present an opportunity for communicable disease transmission when adequate hand hygiene practice and compliance is not adhered to [4,5]. This is because previously conducted studies have established that public restrooms can harbour various communicable pathogens including salmonella and faecal pathogens amongst others [6,7]. These pathogens typically concentrate in wet areas such as around wash hand basins or underneath hand-dryers [8,9,10], or on physical surfaces not properly cleaned [4]. 

Most of the research investigating hand hygiene practice and compliance in community settings has focused on nursery schools, shopping centres or in train stations [11,12,13]. Few studies have focused upon hand hygiene practice and compliance in university settings which is important due to the large numbers of people of various backgrounds interacting daily [14].

Some previous hand hygiene studies conducted in university settings have found compliance to be generally poor [14,15,16,17]. In [16], a hand hygiene compliance rate of 17.4% amongst students was observed. A similar compliance rate of 20% for university students was established by [17], and 22.5% in another similar study by [14]. 

Though efforts have been made in the past to improve hand hygiene practice and compliance, achieving sustained behavioural change is difficult, with most prior intervention strategies having been largely ineffective and expensive [18,19]. This is usually because most hand hygiene interventions are not theory-driven meaning there is not the same level of rigorous scientific evaluation and robustness in addressing the barriers to, and reasons for poor hand hygiene compliance [19,20]. 

One of the most common and affordable interventions used for improving hand hygiene practice and compliance in community settings is via the use of visual posters and signage [15,21,22,23,24,25,26,27]. Most posters and signage act as environmental cues that engage unconscious decision-making processes to prompt behavioural change by focusing on the risk of communicable disease transmission to address poor hand hygiene compliance [28,29,30,31]. 

However, the effectiveness of posters and signage at independently changing behaviour with regards to improving hand hygiene practice and compliance for a sustained length of time is varied and difficult to determine [30]. It is theorised that the effectiveness of many visual cues like posters and signage depends on numerous factors including their design, content, placement, length of placement and target audience [32,33,34,35]. For instance, in [22], hand hygiene compliance increased by 98% after the use of hand hygiene signage. Similarly, in [23], the use of posters and signage lead to an increase of 16.1% on the hand hygiene compliance rate of participants. Similar findings were made in [15] observational study where it was found that the introduction of a visual poster in public restrooms located on a University campus improved the hand hygiene compliance of females significantly more than males. Also, in [26], it was found that the design of the poster used was a significant factor in determining its effectiveness on changing behaviour positively. However, in [21], the use of visual hand hygiene posters did not show any significant improvement in compliance. A similar finding was made in [24] in a veterinary clinic, the presence of hand hygiene posters had no significant effect on improving compliance.

Accordingly, no recent studies have evaluated the impact of a poster intervention on hand hygiene practice and compliance when using public restrooms in a community setting, nor specifically within a university setting. Therefore, the aim of this study was to evaluate the effectiveness of a poster intervention aimed at improving hand hygiene practice and compliance when using public restrooms in a university setting.

## 2. Materials and Methods 

To fulfil the aim of the study, the intervention design took the form of a visual poster (Figure 1). The poster was self-designed and took consideration of how certain behavioural determinants such as the human ‘disgust’ factor can influence hand hygiene practice and compliance as identified in previous studies [13,34]. A dirty handprint showing various microbial growth on hands when not washed was used as the main image of the poster. In addition, one of the main barriers to adequate hand hygiene compliance identified in this study during the pre-intervention observation period was the length of time spent washing and drying hands. In consideration of this, the narrative section of the poster was written so that the mean time it would take to read the poster would be at least the minimum recommended time that should be spent washing hands (20 s) in accordance with the recommended guidelines [36]. 

Research subjects for the study included all University staff, students and visitors who used one set of male and one set of female public restrooms located along the main throughway of a university campus based within the UK. Public restrooms located on a university campus were selected as the research location because it was theorised that they provided the same opportunity for communicable disease transmission when adequate hand hygiene is not adhered too as in other community settings and therefore disease prevention in such a setting must be prioritised.

Each restroom had only liquid soap available for washing hands, and hand dryers for drying hands. The number of cubicles, urinals, sinks and hand dryers in the male (Figure 2) and female restroom (Figure 3) are shown below. A copy of the poster intervention was fixed to each mirror in both restrooms above the sinks at eye level. 

The hand hygiene compliance of the university population when using the public restrooms was measured via indirect observation (no live observer present) using thermal imaging cameras installed above the sink and hand dryer areas in each restroom. The thermal imaging cameras could only observe the sink and hand dryer areas only and were positioned above the entrance to each restroom above head height to remain discreet. In total, it took 90 days to complete the study. This included 30 days of no observation from the date the thermal cameras were first installed. During this time, signage was also erected on the exterior door of each restroom stating that “the sink and hand dryer areas are under observation for research purposes”. This was done in consideration of ethical requirements and as known consent for the study participants. The signage was also erected prior to live observation in consideration of the Hawthorne Effect, to allow research subjects to become accustomed to the presence of the cameras in each restroom. Following this, live observation prior to the installation of the poster interventions in each restroom took place for 30 days between the times of 08:00 and 18:00, Monday to Friday. For the purpose of this study, this period of observation is known as the pre-intervention observation period. After the pre-intervention observation period ended, the poster interventions were installed in each restroom to each mirror over the sinks at eye-level. Further observation was conducted for 30 days between the times of 08:00 and 18:00, Monday to Friday which is otherwise known as the post-intervention observation period. Due to the use of thermal imaging cameras for observation, it was not possible to distinguish whether research subjects were University staff, students or visitors. It was also not possible to identify the number of research subjects who did not use the restrooms under observation due to the signage on the exterior doors. Ethical approval for the project was granted by the university’s ethics filter committee.

All the data collected (thermal video footage) was pre-coded for analysis using a code sheet that recorded relevant hand hygiene behaviours that included:Handwashing compliance (water alone)Handwashing compliance (with soap)Time spent washing hands (>20 s)Hand drying complianceTime spent drying hands (>20 s).

These behaviours were used to classify the hand hygiene compliance of research subjects into four specific categories used in the context of this study including:Adequate hand hygiene compliance involves washing hands with water, soap and then lathering for twenty seconds or more and scrubbing in various rotations and interlocking of fingers, after which hands are rinsed with water to remove soap excess and then dried properly using an appropriate drying method (hand dryer or paper towels) for twenty seconds or more also.Basic hand hygiene involves washing hands with water, soap and scrubbing hands in various rotations and interlocking of fingers after which hands are rinsed with water to remove soap excess, and then dried afterwards using an appropriate drying method (hand dryer or paper towels).Poor hand hygiene involves either washing hands with water alone or not drying hands using an appropriate drying method (hand dryer or paper towels), or both.Non-hand hygiene involves not washing or drying hands.

These hand hygiene categories were self-defined based upon the findings of previous research as well as national guidelines. For instance, good handwashing practice involves using water and soap [37], to scrub the hands efficiently in various rotations including interlocking and interlacing of fingers all over. It is based upon the World Health Organisations ‘5 Moments for Hand Hygiene’ technique, which is a required routine in health care settings [36]. Hand drying is equally important after washing because the transmission of pathogens is more likely to occur from wet skin than dry skin and therefore the mechanical action of drying removes residual pathogens left on wet areas of the skin on hands [10]. The length of time spent washing and drying hands is also significant in the removal of pathogens, with most experts agreeing that twenty seconds is the recommended minimum amount of time that should be spent doing both [37,38,39,40]

### Data Analysis

The data collected was analysed using IBM’s SPSS Statistical Software (v.24) (IBM, New York, NY, USA. Descriptive and inferential statistics were performed. Chi-square analysis was used to identify statistically significant comparisons in the hand hygiene behaviours of the general population when using public restrooms. The statistically significant level was accepted as *p* < 0.05 with Confidence Levels of 95% (CI) reported where applicable. 

## 3. Results

In total, 1149 research subjects (members of the university population) were observed over a 60 day period in the public restrooms selected as the research location. This included 685 research subjects who were observed during the pre-intervention observation period (555 males, 130 females), and 464 research subjects (343 males, 121 females) observed during the post-intervention observation period. Only research subjects who used the public restrooms when under observation were observed, and it was not possible to determine how many research subjects chose not to use the restrooms under observation during both periods. 364 research subjects in total observed the poster during the post-intervention observation period. This included 247 males and 117 females. A summary of the hand hygiene compliance rate of research subjects during the pre- and post-intervention observation periods is presented in Table 1 below. 

Overall, 7.88% of research subjects practiced adequate hand hygiene in the pre-intervention observation period and 7.97% did so during the post-intervention observation. Around 51.09% of research subjects practiced basic compliance in the pre-intervention observation and 55.39% practiced it in the post-intervention observation period. Around 17.37% practiced poor hand hygiene compliance in the pre-intervention observation period and 15.09% did so during the post-intervention period. 23.65% practiced non-hand hygiene compliance during the pre-intervention observation and 21.55% did so during the post-intervention observation period. There was no statistically significant difference between the pre- and post-intervention compliance rate (χ^2^ = 2.38, *p* = 0.50). Gender differences regarding hand hygiene compliance during the pre- and post-intervention observation periods revealed that more females practiced basic hand hygiene in the post-intervention observation period (62.81%) than during the pre-intervention period (49.23%) and this was statistically significant (χ^2^ = 13.49, *p* = < 0.01). There were no significant differences in male hand hygiene compliance for any of the four compliance categories between both observation periods (χ^2^ = 2.24, *p* = 0.52). 

The mean length of time spent washing and drying hands for all research subjects in both the pre- and post-intervention observation periods is shown in Figure 4. During the pre-intervention observation period, the mean length of time spent washing hands by the total number of research subjects was 11.43 s (Std. dev: 9.64). Of these, 11.82% were observed spending 20 s or more washing their hands. The mean length of time spent drying hands was 18.59 s (Std. dev: 10.96) for all research subjects during this period. Regarding gender differences, the mean length of time spent washing hands by males was 11.49 s during the pre-intervention observation period and for hand drying it was 18.40 s. For females, the mean length of time spent washing hands during this period was 11.20 s and 19.29 s for hand drying.

During the post-intervention observation period, the mean length of time spent washing hands by the total number of research subjects was 12.12 s (Std. dev: 9.28). The mean length of time spent drying hands was 20.39 s (Std. dev: 10.63) for all research subjects during this period. The mean length of time spent washing hands for males during this period was 11.71 s and for hand drying it was 20.51 s. The mean length of time spent washing hands for females during this period was 13.03 s and for hand drying it was 20.12 s.

No statistical significance was found for the length of time spent washing hands between the pre- and post-intervention observation periods for all research subjects (χ^2^ = 1.69, *p* = 0.19) nor the time spent drying hands for all research subjects (χ^2^ = 1.18, *p* = 0.28). No statistical significance was found for the length of time spent washing and drying hands for males between both observation periods (χ^2^ = 0.15, *p* = 0.70) nor females (χ^2^ = 3.50, *p* = 0.06).

## 4. Discussion

Despite good hand hygiene practice and compliance being an important preventative measure in communicable disease prevention, the findings of this study indicated that most people in a university setting do not wash or dry their hands adequately. The poor level of hand hygiene compliance found in this study was similar to other studies conducted in university settings [14,15,16,17]. Although visual posters are commonly used to improve hand hygiene practice and compliance, the findings of this study indicated that the use of an independent, self-designed poster to change behaviour and improve hand hygiene practice and compliance was limited in its effect. The reasons for this are complex and may be due to a number of factors. For instance, the limited improvement in overall hand hygiene compliance (adequate and basic hand hygiene) was likely because most visual posters only act as ‘nudges’ or environmental cues that affect behaviour in the moment rather than change behaviour for a sustained length of time [31]. Similar suggestions have been made previously [15,21,30], indicating that single-use poster interventions should not be relied on as an effective form of behavioural change with regards to hand hygiene practice and compliance.

In addition, the limited effectiveness of the posters used in this study at improving hand hygiene practice and compliance may have been because they were self-designed, rather than professionally made like the majority are in community settings, which has been suggested as being a factor in poor hand hygiene compliance in previously conducted studies [15,32]. Also, it is more difficult to change behaviour when an individual does not understand or is not aware of why they are doing it incorrectly, and they may influence and lead others to practicing the same poor method as them. 

Improving the effectiveness of a poster intervention in this setting may be achieved if its content and design are informed by behavioural theory such as the Theory of Planned Behaviour [13]. The Theory of Planned Behaviour can be used to address the barriers to hand hygiene compliance by utilising the human ‘disgust’ factor to change behaviour [13,34]. Also, the length of time spent washing and drying hands was determined as being a key factor in categorising adequate and basic hand hygiene. The number of people that spent at least 20 s washing and drying hands was poor at baseline levels, and only increased slightly overall after the introduction of the poster intervention. However, the difference in the length of time spent washing and drying hands was more apparent for hand drying rather than handwashing. Although not statistically significant, this suggests that the written content of the poster did have a limited, positive effect on the length of time spent washing and drying hands. The greater improvement observed in the length of time spent drying is likely because a traditional hand dryer blows air closer to the 20 s threshold than a sensor tap runs water for in a public restroom due to environmental considerations. It is also likely that the lack of overall improvement in timing is because most people are poor at estimating the length of time they spend washing and drying without having a physical timer present. Therefore, future interventions which focus on improving the length of time spent washing and drying hands should use built environment cues to achieve this. 

Gender differences in hand hygiene practice and compliance indicated that females responded more positively to the poster intervention than males regarding overall hand hygiene practice and compliance. Although it is difficult to determine the reason for this, it may suggest that the design and content of independent hand hygiene posters may influence one gender more so than another. Other, previous studies investigating gender differences in hand hygiene practice and compliance after introduction of visual interventions have made similar findings [33,34]. The reason for the greater improvement in hand hygiene practice and compliance amongst females compared to males after introduction of the poster intervention in this study may have been because females placed greater importance on the efficacy of their hand hygiene practice as a result of having greater awareness of its significance in communicable disease prevention which was a central message of the poster. Males may have been less persuaded by this message because they do not view adequate hand hygiene practice and compliance as a necessary socially acceptable behaviour, nor do they acknowledge its importance in health protection and communicable disease prevention. Similar suggestions have been made by [33] and [34] when investigating gender differences regarding hand hygiene practice and compliance. Built environment factors including the ratio of toilets and urinals to sinks and hand dryers in each male and female restroom may also have influenced hand hygiene practice and compliance in this study. This is because it may have negatively influenced some members of the general population’s intention to practice hand hygiene as they may be unsure whether a hand dryer will be available for drying hands for example, or they felt pressured to dry their hands faster due to the presence of others, which prevents adequate compliance. Hanson et al., 2007 [2] agrees with this view, finding that achieving complete hand hygiene compliance is not possible unless the proper public restroom facilities and amenities are already in place.

Some limitations were identified during the conduct of this study. There may have been systematic error regarding the analysis of the observed data as only one observer (the researcher) was used. In future, two separate analysts will be used to verify the results due to the large volume of footage that was generated. Furthermore, only one set of male and female restrooms were observed due to time, ethical and cost restrictions. In addition, it was not possible to determine whether the people observed each day during both observation periods were not the same members of the university population due to the anonymous nature of the thermal footage recorded. Also, the lower number of females observed compared to males suggests that the Hawthorne Effect may have influenced behaviour. This is because the placement of the signage on the exterior doors of each restroom informing people that the sink and hand dryer areas were under observation for research purposes presumably influenced the number of females that knowingly consented to using the restrooms. 

## 5. Conclusions

This study aimed to assess the effectiveness of a poster intervention on the hand hygiene practice and compliance of a university population when using public restrooms in a university setting. The findings indicated that the hand hygiene practice and compliance of people in this setting are generally poor, and that the use of an independent, self-designed poster intervention has limited effect on improving hand hygiene practice and compliance in this setting, particularly in the short term. The findings also demonstrate the usefulness of using thermal imaging cameras for monitoring hand hygiene practice and compliance, as a large number of people can be observed simultaneously and a range of behaviours can be reliably recorded. Future research should focus on exploring the reasons for the limited effectiveness of a self-designed poster intervention on hand hygiene practice and compliance in a university setting, and how behavioural determinants including the presence of others affect compliance. Future research should also investigate whether a range of poster interventions that are informed by behavioural theory can successfully change behaviour, and further explore the usefulness of thermal imaging cameras for monitoring hand hygiene behaviour in the wider community setting. 

## Figures and Tables

**Figure 1 ijerph-16-05036-f001:**
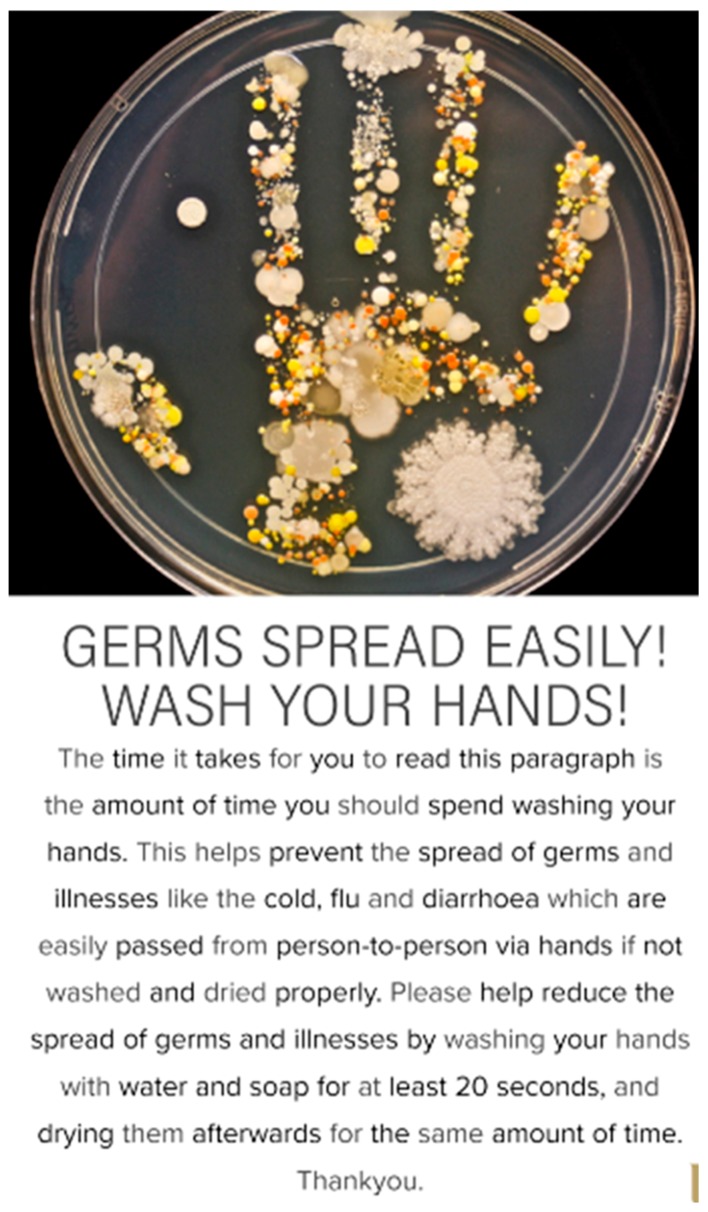
Informed intervention designed as a visual hand hygiene poster.

**Figure 2 ijerph-16-05036-f002:**
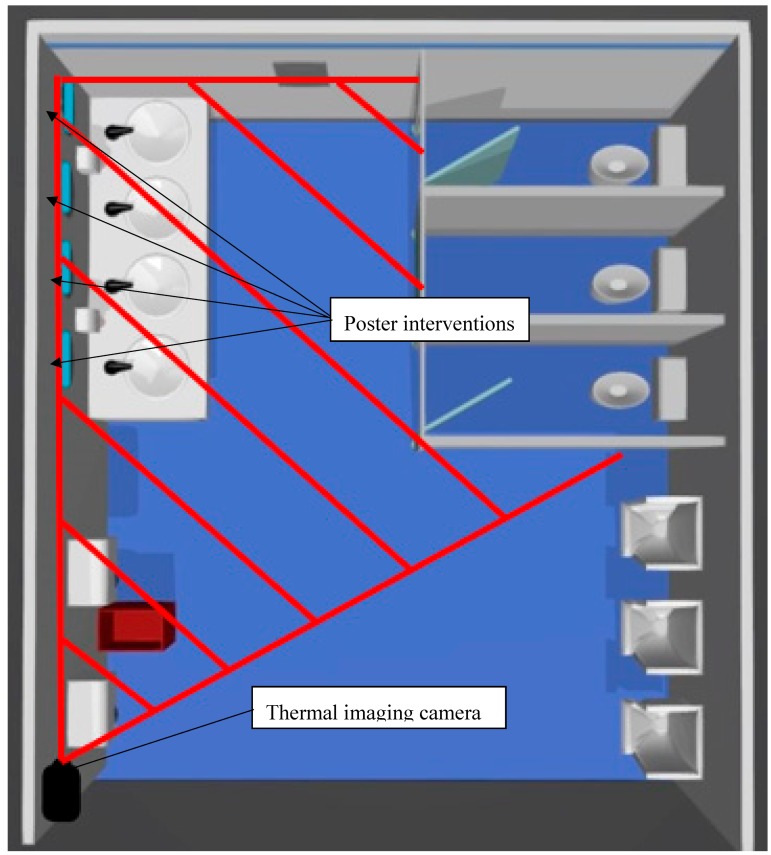
Model of the male public restrooms.

**Figure 3 ijerph-16-05036-f003:**
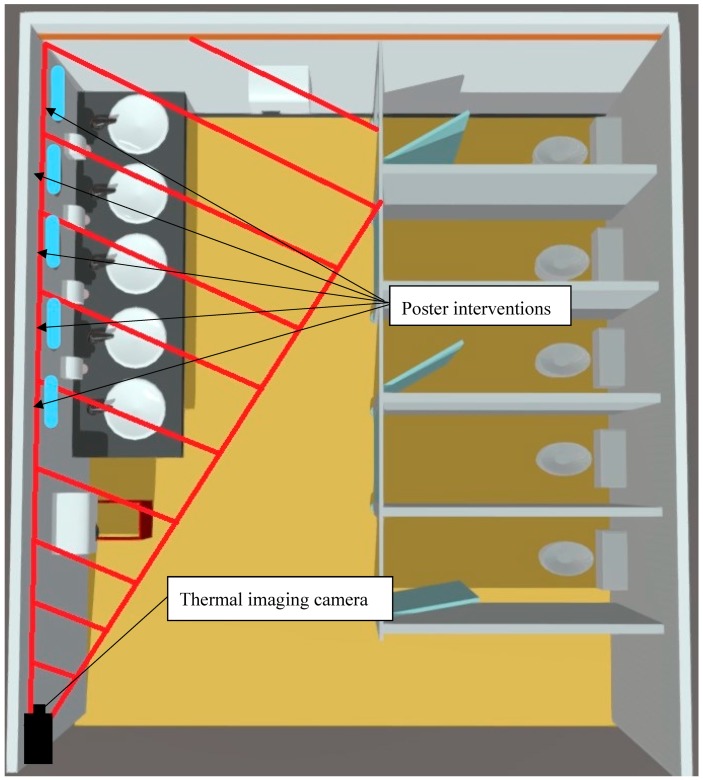
Model of the female public restrooms.

**Figure 4 ijerph-16-05036-f004:**
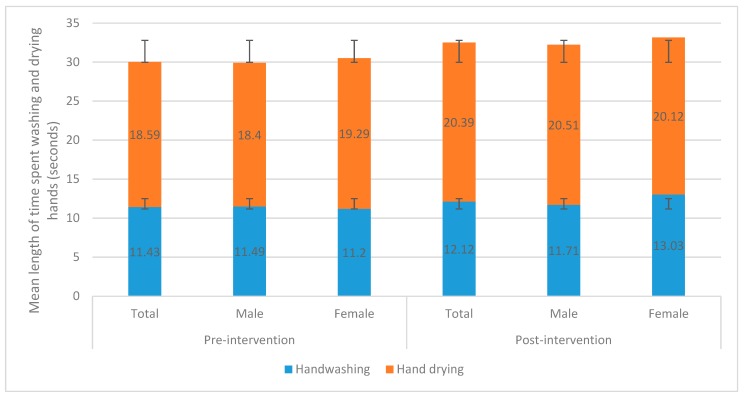
Length of time spent washing and drying hands during the pre- and post-intervention observation periods for the total number of research subjects and by gender.

**Table 1 ijerph-16-05036-t001:** Hand hygiene compliance rate during the pre- and post-intervention observation periods.

Hand Hygiene (HH) Compliance Category	Total Hand Hygiene Compliance Rate by Observation Period *n* (%)					
	Pre-Intervention			Post-Intervention		
	Total	Male	Female	Total	Male	Female
Adequate HH	54 (7.88)	46 (8.29)	8 (6.15)	37 (7.97)	26 (7.58)	11 (9.09)
Basic HH	350 (51.09)	286 (51.53)	64 (49.23)	257 (55.39)	181 (52.77)	76 (62.81)
Poor HH	119 (17.37)	82 (14.77)	37 (28.46)	70 (15.09)	40 (11.66)	30 (24.79)
Non-HH	162 (23.65)	141 (25.41)	21 (16.15)	100 (21.55)	96 (27.99)	4 (3.31)
Total	685 (100.00)	555 (100.00)	130 (100.00)	464 (100.00)	343 (100.00)	121 (100.00)

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
