# Peer review of "The Effectiveness of a Poster Intervention on Hand Hygiene Practice and Compliance When Using Public Restrooms in a University Setting"

_ijerph, 2019, doi:10.3390/ijerph16245036_

Round 1

Reviewer 1 Report

Comments

Effectiveness of a poster intervention on the hand hygiene compliance of the general population when public restrooms

This study described a poster intervention was designed and installed in a set of male and female public restrooms located on a local University campus. The authors evaluated the effectiveness of this poster intervention. The issue on searching washing hand intervention is interesting, but the serious problem in this study is that it didn’t know how many university students have watched the poster. Therefore, it is difficult to say the observed subjects washed their hand due to the poster’s effectiveness although the poster was installed.  

There are some further comments as the followed:

The English language is confusing in several parts of the manuscript and a careful review is recommended.

1. Introduction

It is recommended to add content to make the last three paragraphs in this part be further optimized so that to enhance the logic of the background.

2. Materials and Methods

1) What is the source of the poster? Was it based on reference or self-designed? I am afraid the scope of application of the study is narrow for the only one self-designed poster.

2) There is no explanation of the basis of hand hygiene behavior categories. A more careful description of the materials and methods is welcome, for example the rationality of using 20 seconds for both dry phones and paper towels, and whether the use of hand dryer can cause secondary contamination of hands with bacteria and mold should be taken into consideration. This section should be revisited and updated with appropriate references to ensure it reflects currently research and understanding of this process.

3) The study was conducted in a university campus. Were the research subjects mainly composed of students? The sample representation is questionable.

3. Results

1) The large difference in the number of men and women observed is confusing and should be clarified.

2) The total figure “130” in the last column of table 1 is incorrect, it should be 464.

3) Did you observe how many people noticed and read the poster?

4) The title and content of Figure 2 do not correspond which would confuse readers, the overall length of time is not shown.

4. Discussion

1)    Please check line 166-168. The presentation is inconsistent with the result, we didn’t see the result about the “small increase in adequate hand hygiene compliance”.

Reviewer 2 Report

I think this is a really important topic since one of the most common ways of promoting handwashing is through the use of simple posters. I think this article has the potential to challenge some of the common assumptions about handwashing behaviour change.

Abstract:

Since you are writing for a global audience remove the term ‘local’ on line 14 and state the country.

Presumably the study did not explore the behaviour of the ‘general population’ but rather the behaviour of students and staff at the university.

Intro:

In your review of the literature I would consider including more information on work around visual cues (since this is also basically what you are testing). Examples include:

Judah, G., et al. (2009). "Experimental pretesting of hand-washing interventions in a natural setting." American journal of public health 99 Suppl 2: S405-411. Dreibelbis, R., et al. (2016). "Behavior Change without Behavior Change Communication: Nudging Handwashing among Primary School Students in Bangladesh." Int J Environ Res Public Health 13(1). Pfattheicher, S., et al. (2018). "A field study on watching eyes and hand hygiene compliance in a public restroom." Journal of Applied Social Psychology 48(4): 188-194. Johnson, H. D., et al. (2003). "Sex differences in public restroom handwashing behavior associated with visual behavior prompts." Perceptual & Motor Skills 97(3 Pt 1): 805-810.

These studies also could be useful references since they are done at a university and describes the influence of the physical environment:

Mariwah, S., et al. (2012). "The impact of gender and physical environment on the handwashing behaviour of university students in Ghana." Trop Med Int Health 17(4): 447-454. Anderson, J. L., et al. (2008). "Gender and ethnic differences in hand hygiene practices among college students." Am J Infect Control 36(5): 361-368.

You also refer to several texts that took place in hospital settings. While this is useful you may want to reflect on whether these institutional and higher risk settings are a good reflections of what the ‘general public’ do when outside these settings.

I feel the statement on line 58 is inaccurate given some of the texts above and I don’t think it necessarily makes the case for your researcher stronger.

I think that generally the intro lacks structure. I would consider restructuring the paragraphs to emphasise the following points:

Public health relevance of handwashing and rates of handwashing globally Common approaches to changing handwashing behaviour in community gettings/among general public. Emphasising that one of the most common and affordable approaches is the delivery of handwashing promotion through posters or visual cues. Summarise the research on visual cues. Highlight the gaps in the literature and how your work will contribute to this.

In the Intro you may also want to highlight that a strength of your study is that it is looking at the independent effect of a poster and whether this alone is sufficient to change behaviour. This is really useful to flag to the reader since posters are often used as one component of more complex interventions.

In the methods, or potentially in the intro, I would provide a deeper discussion of the research setting. Ie where in the university were the toilets located? How many cubicles/handwashing facilities were there? What type of soap and handwashing facilities were available? Also provide a brief summary about why you chose to test in a university setting.

Were there any public notifications about the study? Were participants aware of the observation?

Did you look at whether the presence of other people in the bathroom increased handwashing rates? This has been noted in several other studies and could be interesting to include. Ie. Perhaps posters work better if there is an element of social judgement.

The handwashing durations you observed are actually higher than reported in other studies. E.g. Garbutt, C., et al. (2007). "The public hand hygiene practices of New Zealanders: a national survey." N Z Med J 120(1265): U2810. It might be interesting to reflect on why this might be the case.

I would also be careful not to place too much emphasis on the ’20 seconds’. Yes this is in some guidelines (other guidelines recommend 40-60 seconds). It is basically an arbitrary number. There is some argument to be made that washing hand more frequently for a shorter duration may actually be more effective in reducing disease. Indeed a lot of pathogens are removed after about 10 seconds. While there is no harm in encouraging people to wash their hands for longer too much focus on this could actually make handwashing seem less feasible to practice regularly.

The statement on line 211 and 212 is unclear. What biases (do you possibly mean design choices?)? How could these have been mitigated? What other options could be worth exploring?

You also mention observer bias. However I don’t think this is what you mean. Observer bias is basically the same as the hawthorne effect. I think you are actually talking about systematic error (that is that there may have been errors in the data analysis given the amount of footage). If this is the case then maybe you could recommend that future studies using this approach make use of two separate analysists. You then do mention the hawthorne effect but it is not really clear how this occurred. To what extent could bathroom users see the camera – this should actually be documented in the methods.

I feel that some of the points in your discussions and limitations sections feel a bit porceedural. You raise a potential issue and dismiss it. For example, I think it’s unlikely that placement of the posters was an issue here, given your explanation in the methods. There is no where else more logical to put them. So I would drop this as it is unlikely to be an explanation for your results. You also talk about the duration of the poster being up. Again I find this an implausible rationale as your poster has a simple theory of change and does not aim to change behaviour over the long term but rather acts as an in-the-moment cue. The same applies to your discussion of bias – only discuss things that are likely to have had a genuine impact and explain what this impact is likely to have been.

I would suggest including some reflections on whether a poster may be more effective if linked to a range of other behaviour change components. I would also suggest reflecting on how the dynamics of the university environment may have affected your results and the implications of this for generalisability.

Your conclusion seems to kind of contradict your discussion and present an overly rosy view of the results.

Reviewer 3 Report

The manuscript reports on the evaluation of a poster intervention to incentivise handwashing. Specifically, the authors document the failure of a poster placed in public restrooms on a university campus to improve observed handwashing behaviours and discuss some differences between male and female users. The manuscript is clearly written. However, little information is provided on the poster development process, results could be presented more efficiently, and the scope of the study remains limited to a specific, likely educated population and a specific intervention, with lessons learned that are difficult to generalise.

Title

There is an issue with the title (missing word?) and the names of authors appearing in the title too in the PDF that I received.

Abstract

Line 15: indirect “observations”?

Introduction

Posters are “nudges” – some background/theory on how this type of intervention works could be included in the introduction (see Caris et al., 2018, J. Hosp. Inf. and Grover et al., 2017, TMIH, for a recent review and example of nudges to promote handwashing).

Methods

More details are needed regarding poster development. In introduction, the authors mention factors influencing the effectiveness of a poster intervention (lines 58-59) but it is unclear from the methods how these factors were taken into account in the development of the poster used for this study. In particular, the authors should address why they thought the contents were appropriate for their target audience and whether any pre-testing/piloting was conducted to evaluate the poster.

Line 72-73: were participants informed that they would be filmed while using the restrooms? Details on how information about the study was shared with restroom users, whether consent was obtained, etc. would be useful. This could be placed at the end of the methods section along with the ethics approval information.

Line 80-81: it is currently unclear how many restrooms were targeted, and how large they were (number of sinks and mirrors in each restroom, or even a map showing where the poster was placed).  

Results

The three graphs could easily be presented as one. Likewise, Table 1 and Table 2 could be grouped.

Line 111: any explanation regarding why so many more male than female users were observed? For example, if the male restrooms were larger, is it possible that the poster was less likely to be visible to users? To add in discussion.

It seems like by observing handwashing on camera footages, the specific time spent washing hands by each user could be recorded. Presenting/analysing this as a continuous outcome may provide more insight into handwashing behaviours than using a binary outcome (e.g. how long do people who wash hands spend washing and drying? How does this compare to the recommended 20 seconds?).

Discussion

The authors should highlight what their contribution is to the existing literature (for example, to my knowledge, the use of thermal cameras to monitor handwashing is a novel approach) and discuss how lessons learned may be applied to their university campus or other settings. Specificities of their study population could also be discussed.

Conducting focus groups with the target population to get feedback on the poster design would be a relatively easy addition to this research to understand better why the poster had a limited impact.

Line 217: mention of the Hawthorne effect suggests that users knew they were filmed – this should be clear from the methods. And given the observed results, how valid is this concern?

Conclusions

Line 225: “the design, content and placement of a poster are critical…” – this conclusion is not supported by the research, since only one poster was tested, placed in the same place in each restroom.

Line 226: given that even short-term impact could not be demonstrated in this study, do the authors really believe that investigating the long-term impacts of similar interventions is necessary?

Round 2

Reviewer 1 Report

The revised version is much better than the previous one. However,the whole text is not so scientific and rigorous. There is still some parts should be treated.

In Abstract: 

Line 20-21:Handwashing and hand drying timings and gender-related results were also recorded.

This sentence is not the result. It is should move to the methods section.

Line 23: Female hand hygiene compliance improved more than males but not significantly.

We cannot find such a result in the ABSTRACT section.

Author Response

In Abstract: 

Line 20-21:Handwashing and hand drying timings and gender-related results were also recorded.

This sentence is not the result. It is should move to the methods section.

This has been removed from the abstract and this variable is detailed in the methodology section.

Line 23: Female hand hygiene compliance improved more than males but not significantly.

We cannot find such a result in the ABSTRACT section.

This has been removed and instead in the results section of the abstract, basic hand hygiene compliance significantly improved between the pre and post-intervention observation periods.

Reviewer 3 Report

The introduction and methods have substantially improved and now provide a good overview of the context and processes involved in this research. 

In Results & Discussion:

Lines 195-196, 199-200: it is unclear which groups were compared to determine statistical significance? 

Figures 4-6: as mentioned in my previous review, these graphs could easily be grouped into one. One bar could show washing & drying times together (so that it illustrates the total time spent by users on hand hygiene). Graphs should also include error bars (standard deviations). 

Lines 281-285: the explanation provided by the authors in response to comments (different numbers of toilets/urinals and sinks in female and male restrooms) is sensible and would be worth mentioning in the manuscript as well.  

Author Response

Lines 195-196, 199-200: it is unclear which groups were compared to determine statistical significance? 

All research subjects in the pre-intervention group and all research subjects in the post-intervention observation group were compared to determine statistical significance. This has been amended in the revised version.

Figures 4-6: as mentioned in my previous review, these graphs could easily be grouped into one. One bar could show washing & drying times together (so that it illustrates the total time spent by users on hand hygiene). Graphs should also include error bars (standard deviations). 

These amendments to the figures have been made.

Lines 281-285: the explanation provided by the authors in response to comments (different numbers of toilets/urinals and sinks in female and male restrooms) is sensible and would be worth mentioning in the manuscript as well.  

This has been included in the discussion in the section regarding gender differences.